# LLM4CTI: Uncovering Security Entities and Their Interactions from Unstructured Cyber Threat Intelligence Articles

## Abstract

Unstructured Cyber Threat Intelligence (CTI) articles are a critical source of detailed threat knowledge. Structuring this information into knowledge graphs is crucial for downstream applications, including enhancing Large Language Models (LLMs) factuality and enabling advanced predictive analysis. However, extracting security-related knowledge from these long articles poses a fundamental dilemma for LLMs: capturing relationships scattered across distant sentences requires full-article context, yet processing long inputs triggers issues like the "lost in the middle" phenomenon, while naive chunking severs these crucial long-range dependencies. To resolve this dilemma, we present LLM4CTI, an approach that employs a novel dual-context, chunk-wise processing pipeline. For each text chunk, LLM4CTI provides the LLM with both the local chunk for fine-grained analysis and the full article for global context. A formal, security-oriented framework further guides the LLM to focus on threat-relevant information. Evaluation shows that LLM4CTI achieves a state-of-the-art F1-score of $85.13\%$. This is driven by an average recall of $81.58\%$, substantially outperforming the recall of competitors like CTINexus ($65.97\%$), CTIKG ($76.73\%$) and GraphRAG ($70.25\%$), while maintaining a high precision of $89.87\%$. Notably, the version of LLM4CTI powered by a local 32B open-source model also achieves a strong F1-score of $81.58\%$, surpassing baselines that rely on the much larger GPT-4o. Furthermore, LLM4CTI's integrated GNN leverages the extracted knowledge to deliver predictive insights. It successfully forecasts relationships between threat-related entities, such as connections between malware families, with over 80% Hits@10 accuracy.

## 1 Introduction

Large enterprises are increasingly being plagued by cyber threats (Gao et al., 2018; Hutchins et al., 2011; dep, 2021; Times, 2014). To mitigate these threats, they actively collect and share Cyber Threat Intelligence (CTI) (McMillan, 2013; Wagner et al., 2019), enabling both human analysts and automated systems to respond swiftly. CTI includes Indicators of Compromise (IOCs) (Obrst et al., 2012; Senki, 2016), Common Vulnerabilities and Exposures (CVEs) (MITRE, 2020), and cyber kill chains (cyb, 2021; Corporation, 2022). Beyond structured CTI feeds, articles and posts from blogs and forums, referred to as *CTI articles*, offer valuable intelligence by describing real-world attacks in greater detail than IOCs, including behaviors critical for threat investigation.

Given the importance of CTI, prior work (Li et al., 2022; Satvat et al., 2021; Gao et al., 2022) has focused on extracting structured knowledge from CTI articles in threat knowledge bases like MITRE ATT&CK (Corporation, 2022) and the NVD (of Standards & Technology, 2021). Recent advances in Large Language Models (LLMs)(Wang et al., 2023; OpenAI, 2023) have enabled new approaches for automated CTI knowledge graph construction, such as CTINexus(Cheng et al., 2024), CTIKG (Huang & Xiao, 2024), and AttackG+(Zhang et al., 2024). These approaches extract security entities (e.g., CVEs, malware, threat actors) and their interactions (e.g., malware evolution) from diverse CTI texts, capturing richer threat knowledge beyond traditional sources. For example, Figure 1 shows how different CTI articles report distinct incidents involving Log4Shell (CVE-2021-44228), including EnemyBot and BugWorm. Structured knowledge graphs are essential for advanced security tasks, including mapping low-level system events to high-level threat behav-

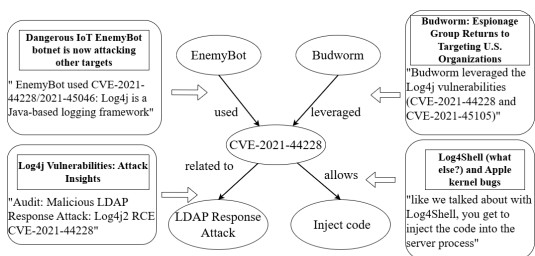

Summary:
CVE-2021-44228, commonly known as Log4Shell, is a critical vulnerability in Apache's Log4j logging library that allows attackers to execute code remotely on vulnerable systems. By exploiting this flaw, attackers can bypass authentication protocols, use code injection, and control affected systems. This vulnerability has been used by groups like Budworm and EnemyBot. Budworm targets U.S. organizations and the defense sector, leveraging Log4Shell for attacks via LDAP protocols and DLL side-loading. EnemyBot, on the other hand, uses it to compromise IoT devices and CMS servers, which often lack strong security controls.

Figure 1: CTI articles for Log4Shell (CVE-2021-44228)

iors (Milajerdi et al., 2019) (Appendix A ) and predicting semantic links to reveal hidden patterns such as malware lineage, enabling proactive defense.

While these approaches achieved some early promising results, extracting CTI knowledge from long cyber security-related texts and constructing knowledge graphs still faces several challenges. ❶: Security entity behaviors are often dispersed across multiple sentences. Yet, when provided with lengthy contexts, LLMs may display **laziness** (Zhao et al., 2024), while naive chunking strategies disrupt cross-chunk coherence, creating an inescapable dilemma. ❷: CTI articles are frequently padded with irrelevant content such as advertisements, sidebars, and headers. For an LLM, critical security entities and their relationships become information buried deep within a long context, making them susceptible to the **"lost in the middle" phenomenon** (Liu et al., 2024). ❸: CTI articles often introduce threat entities early and describe their actions later, requiring robust coreference resolution. While LLMs can trace such links, doing so across long documents demands lengthy reasoning chains that exceed token limits and frequently result in **hallucination** (Huang et al., 2025), compromising output reliability.

In this paper, we propose LLM4CTI, *an LLM-based approach that (1) automatically extracts threat intelligence from a wide range of CTI articles and structures it into a knowledge graph, and (2) leverages this graph to uncover semantic relationships among security entities (e.g., CVEs or malware) and their interactions.* In particular, LLM4CTI has the following innovative designs to address the above challenges:

- **Chunk-wise Processing with Dual Context**: To handle long articles and mitigate hallucination, laziness, and lost in the middle problem. LLM4CTI segments articles into smaller chunks. For each chunk, it provides the LLM with two contexts: the local chunk for detailed analysis and the full article for global context. This dual-context strategy enables the model to capture both immediate details and long-range dependencies.
- **Formal Framework for Guided Extraction**: Inspired by STIX 2.1 (TC, 2017) but extended for practical use, we introduce a formal framework to define security entities and their relationships. This framework guides the LLM to focus specifically on threat-relevant information. By first identifying entities and *then* their interactions, our approach improves the quality of extracted knowledge and filters out irrelevant text.
- **Step-wise Knowledge Extraction Pipeline**: LLM4CTI employs a structured, multi-step pipeline for each text chunk. It sequentially (1) identifies threat-related entities based on our formal framework, (2) extracts relationships *within* the chunk, and (3) infers relationships *across* different chunks. This guided, step-by-step process ensures a more comprehensive extraction of security knowledge.
- **GNN-based Semantic Relationship Prediction**: After constructing the knowledge graph, LLM4CTI uses it to train a Graph Neural Network (GNN). This model learns to predict the like-

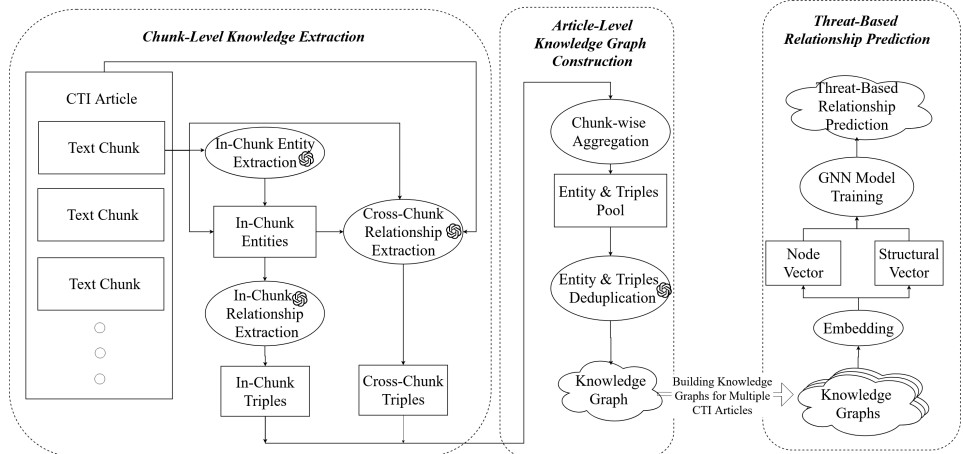

Figure 2: Architecture of LLM4CTI

lihood of relationships between security entities, even for emerging threats not explicitly linked in the source articles. This predictive capability helps analysts uncover latent connections and proactively defend against new attacks.

**Evaluations**. We collect $88,803$ CTI articles from $67$ online sources, including $672$ ATT&CK articles (Corporation, 2022), to construct our CTI corpus. Based on this corpus, we curate two evaluation datasets: (1) the *Diverse CTI Dataset*, comprising 55 articles with a ground truth of 791 entities and 744 relationships to assess knowledge graph construction quality; and (2) the *Top Security Entity Dataset*, containing 302 articles that form a graph with 11,494 nodes and 14,972 edges to evaluate downstream link prediction.

Our evaluation on the Diverse CTI Dataset demonstrates that LLM4CTI significantly outperforms state-of-the-art baselines in knowledge graph construction. The LLM4CTI powered by o4mini leads all systems, achieving a precision of 89.87% and a recall of 81.58%, resulting in a top F1-score of 85.13%. The variant of LLM4CTI powered by the open-source QWQ 32B model shows excellent performance, with a precision of 85.58%, recall of 78.66%, and an F1-score of 81.58%. This advantage is particularly evident in longer articles. Because of its detailed prompt design, chunking strategy and global context view, LLM4CTI maintains high recall on documents exceeding 1,500 tokens, whereas the recall of single-pass systems like GraphRAG and CTINexus drops sharply.

Notably, our design enables this QWQ 32B-based approach, running on a local GPU, to outperform baselines that rely on the significantly larger GPT-4o model, such as CTIKG (78.98% F1), CTINexus (75.27% F1) and GraphRAG (79.16% F1), while offering superior cost-efficiency on both cloud and local GPU deployments.

Furthermore, we demonstrate the practical utility of the generated knowledge graphs in a downstream link prediction task (RQ3). The results show that the graph provides powerful predictive intelligence for security analysts. For example, when presented with a new entity, the system can infer its likely relationships to existing entities with high accuracy, achieving a Hits@10 rate of 80.57% for predicting connections between malware families and 78.15% for linking malware to threat actors. This capability can significantly accelerate threat investigation by providing a ranked list. The codes and datasets of LLM4CTI are available at our project website: https://github.com/hiddenauthor/LLM4CTI. (cti, 2025).

## 2 OVERVIEW

Figure 2 illustrates the overall architecture of LLM4CTI, consisting of three phases: chunk-level knowledge extraction, article-level knowledge graph construction, and Threat-Based Relationship Prediction.

**Chunk-Level Knowledge Extraction.** Given a CTI article, LLM4CTI first segments it into semantically coherent text chunks. For each chunk, LLM4CTI employs an LLM agent to perform in-chunk entity extraction that identifies cyber threat-related entities and assigns pre-defined types. Then, LLM4CTI takes the whole CTI article content as the background information, and instructs another LLM agent to perform both in-chunk relationship extraction and cross-chunk relationship extraction, generating in-chunk triples and cross-chunk triples, respectively.

**Article-level knowledge graph construction.** After processing each chunk, LLM4CTI sequentially stores the extracted entities and triples in an entity & triple pool, preserving the boundaries between chunks. LLM4CTI then employs a dedicated LLM agent to perform entity and triple deduplication in the pool and produce an article-level knowledge graph.

**Threat-Based Relationship Prediction.** In this final phase, LLM4CTI utilizes the constructed knowledge graphs for its primary application. To predict potential relationships between entities, it generates text embeddings for the nodes and edges in a graph. These embeddings are then used to train a GNN model, which can determine whether a relationship likely exists between a given pair of entities.

## 3 APPROACH

This section defines knowledge graph structure of LLM4CTI and describes its major phases.

### 3.1 FORMAL STRUCTURE OF KNOWLEDGE GRAPH

A cyber threat knowledge graph is a semantic structure of *triples*, each representing a fact extracted from unstructured CTI texts. Each *triple* includes two typed *entities* and a directed *relationship* linking them. Formally, an `entity` refers to any unit of information that can be independently identified, described, and analyzed in the context of cyber threat intelligence. A **relationship** defines the semantic connection between two entities, captured by a surface-level phrase (`rel`) and a mandatory, normalized category (`rel_type`). A triple is a 3-tuple: ⟨sub, rel, obj⟩. It connects two entities via a directed relationship and serves as the core structure for knowledge extraction and graph construction. More details for the formal representation are provided in Appendix B.

### 3.2 CHUNK-LEVEL KNOWLEDGE EXTRACTION

Processing long CTI articles poses challenges for LLMs, which tend to underperform on long inputs due to issues like the lost in the middle and hallucinations (Huang et al., 2025), and practical output token limits. To mitigate these problems, LLM4CTI first segments each article into smaller, semantically coherent text chunks.

Inspired by RAG practices (Juvekar & Purwar, 2024). LLM4CTI recursively segment articles into semantic chunks of up to 400 tokens with a 10% overlap, prioritizing splits at paragraph and sentence boundaries.

For each chunk of a CTI article, LLM4CTI performs structured knowledge extraction using a prompt-driven LLM agent. To support accurate and context-aware extraction, the input to the agent is not limited to the chunk itself; instead, both the chunk and the whole article content are provided as input. This dual-input design allows the LLM to recognize the local context of the chunk while retaining awareness of the article's global topic, enabling more accurate identification of implicit relationships of entities. Figure 3a provides an illustration of this prompt, which is described in further detail below.

**In-Chunk Entity Extraction**. The prompt's entity extraction component identifies entities relevant to cyber threats. The complete prompt for single-chunk entity and relationship extraction is shown in Appendix C. The prompt of entity extraction section involves:

- **Entity Definition & Entity Extraction Rules.** Both explicit mentions (e.g., Emotet) and implicit references (e.g., "RAR archive", or pronouns such as "it") are extracted, as long as they are independently identifiable and contextually meaningful within the chunk. To resolve pronouns, the

| Prompt Section | |
|---|---|
| Global Goal | |
| | Entity Definition |
| | Entity Extraction Rules |
| In-Chunk Entity Extraction | Normalization Rules |
| | Type Assignment Rules |
| | Alias & Parent Entity Rules |
| | Predefined Entity Types |
| | Relationship Definition |
| In-Chunk Relationship Extraction and Cross-Chunk Relationship Extraction | In-chunk Relationship Extraction Rules |
| | Cross-chunk Relationship Extraction Rules |
| | Predefined Relationship Types |
| Output Format Specification and Example | |
| Actual Input: < Article Full Text >, < Current Chunk Text > | |

(a) Prompt for chunk-level knowledge extraction

| Prompt Section | |
|---|---|
| Global Goal | |
| | Entity Deduplication Rules |
| Entity Deduplication | Alias Name Handling |
| | Parent Entity Handling |
| | Triples Deduplication Rules |
| Triples Deduplication | Implicit Link Completion |
| | Merging Integrity Constraints |
| Output Format Specification and Example | |
| Actual Input: <Index of chunk 1>, <Entity list from chunk 1>, <Relationship from chunk 1>, <Index of chunk 2>, <Entity list from chunk 2>, <Relationship from chunk 2>... | |

(b) Prompt for knowledge graph construction

Figure 3: LLM4CTI's prompt structures for (a) chunk-level knowledge extraction and (b) knowledge graph construction.

model performs coreference resolution using both the chunk under analysis and the full article context.

- **Type Assignment Rules.** Each entity is assigned exactly one type from the predefined taxonomy introduced earlier. Unclassified entities are labeled as *other*.
- **Alias & Parent Entity Rules.** If an entity has multiple surface forms or is a subcomponent of another, this information is recorded in the *alias* and *parent entity* fields, respectively.
- **Normalization Rules.** Obfuscated indicators are standardized to canonical forms.
- **Self-reflective Validation.** The model conducts a post-extraction check to identify potentially missed entities based on contextual cues.

**In-Chunk Relationship Extraction**. After identifying all relevant entities within a chunk, LLM4CTI extracts both explicit and implicit relationships among them. For each relationship, LLM4CTI records a surface-level expression (*rel*) extracted from the chunk, along with one or more normalized relation types (*rel_type*) from a predefined taxonomy. The predefined types also guide the identification of implicit relationships conveyed through relative clauses, modifiers, or indirect linguistic patterns. These extracted elements are combined to form structured `triples`.

**Cross-Chunk Relationship Extraction**. LLM4CTI extracts both direct and indirect references as triples, linking entities across chunks via explicit mentions or implied cues like pronouns or structure. This preserves continuity in the knowledge graph. To minimize isolated entities, LLM4CTI examines broader context, especially the main threat and list-based indicators, to infer hidden links. These inferred triples integrate IOCs and other entities that might otherwise remain disconnected. A final validation step confirms all entities are linked within or across chunks.

### 3.3 KNOWLEDGE GRAPH CONSTRUCTION

**Chunk-wise Aggregation**. After completing entity and relationship extraction for each chunk, LLM4CTI aggregates all results into the entity & triples pool. This pool is a sequential list that preserves chunk boundaries, where each entry contains the extracted entities and triples from a single chunk. Each block is marked by the start and end positions of the corresponding chunk and serves as the input for subsequent deduplication and graph construction steps.

**Entity and Triples Deduplication**. After extracting entities and triples from each chunk, LLM4CTI applies an LLM-based merger agent to consolidate them into a unified article-level knowledge graph. Figure 3b presents the prompt structure for article-level merging. The full prompt

Table 1: Comparison of knowledge graph construction

| Article Topic | LLM4CTI (QWQ 32B) | | | LLM4CTI (OpenAI o4-mini) | | | CTINexus | | | GraphRAG | | | Extractor | | | CTIKG | | |
|---|---|---|---|---|---|---|---|---|---|---|---|---|---|---|---|---|---|---|
| | Precision | Recall | F1 | Precision | Recall | F1 | Precision | Recall | F1 | Precision | Recall | F1 | Precision | Recall | F1 | Precision | Recall | F1 |
| APT | 90.75% | 80.77% | 85.47% | 92.39% | **95.71%** | **94.02%** | 92.56% | 75.68% | 83.27% | **94.65%** | 76.90% | 84.86% | 24.76% | 64.88% | 35.84% | 83.55% | 90.85% | 87.04% |
| Adware | 85.77% | **91.67%** | 88.62% | 91.74% | 91.00% | **91.37%** | 91.61% | 84.50% | 87.91% | **98.82%** | 77.17% | 86.66% | 16.92% | 73.67% | 27.52% | 83.40% | 91.00% | 87.03% |
| Botnets | 77.52% | 77.57% | 77.54% | 91.63% | **84.90%** | **88.14%** | 84.84% | 74.32% | 79.23% | **95.80%** | 63.96% | 76.70% | 19.13% | 37.63% | 25.36% | 87.02% | 81.18% | 84.00% |
| DoS attack | 83.76% | 78.57% | 81.08% | 87.41% | 52.38% | 65.51% | 88.80% | 52.38% | 65.89% | 82.96% | **95.24%** | **88.68%** | 6.94% | 19.05% | 10.18% | **95.00%** | 33.33% | 49.35% |
| Ransomware | 90.63% | 73.57% | 81.21% | 76.75% | 75.20% | 75.97% | **95.66%** | 67.93% | 79.45% | 91.33% | 50.53% | 65.06% | 35.15% | 54.11% | 42.61% | 82.61% | **86.02%** | **84.28%** |
| Spam/Phishing | 86.58% | 66.69% | 75.34% | 85.01% | 70.51% | **77.08%** | 92.92% | 56.59% | 70.34% | **93.52%** | 57.57% | 71.27% | 17.78% | 30.37% | 22.43% | 81.89% | **71.88%** | 76.56% |
| Supply Chain Attacks | 79.91% | **95.38%** | 86.96% | 90.15% | 85.13% | 87.57% | 86.39% | 68.21% | 76.23% | **91.06%** | 85.90% | **88.40%** | 14.00% | 36.67% | 20.27% | 81.15% | 68.72% | 74.42% |
| Trojan | 85.05% | 63.18% | 72.50% | **98.46%** | 85.89% | **91.75%** | 73.32% | 45.84% | 56.41% | 97.92% | 58.58% | 73.31% | 14.13% | 37.87% | 20.58% | 79.17% | 64.53% | 71.10% |
| Virus/Worm | 91.69% | 79.36% | 85.08% | 92.06% | 89.91% | **90.97%** | 94.31% | 69.66% | 80.14% | 94.18% | 75.15% | 83.59% | 25.05% | 52.54% | 33.92% | 82.60% | **92.67%** | 87.34% |
| Vuln. & exploits | 84.17% | 79.84% | 81.95% | **93.07%** | 85.19% | **88.96%** | 86.10% | 64.58% | 73.80% | 89.87% | 61.49% | 73.02% | 23.19% | 52.73% | 32.21% | 90.33% | **87.14%** | 88.71% |
| Average (macro) | 85.58% | 78.66% | 81.58% | 89.87% | 81.58% | **85.13%** | 88.65% | 65.97% | 75.27% | **93.01%** | 70.25% | 79.16% | 19.70% | 45.95% | 27.09% | 84.67% | 76.73% | 78.98% |

is provided in Appendix D. The deduplication process handling semantic equivalence and alias for both entities and triples (detailed in Appendix E).

**Multi-Article Aggregation**. To build a unified knowledge graph from multiple articles, LLM4CTI aggregates the extracted triples from each article-level graph into a global graph. If a node already exists in the global graph, its attributes are merged by appending new values to existing ones and removing duplicates. LLM4CTI constructs a multi-edge graph with source-annotated relations and then uses spaCy (Explosion AI, 2024) to lemmatize and merge semantically redundant nodes.

### 3.4 SEMANTIC RELATIONSHIP PREDICTION

To predict relationships, LLM4CTI employs a GNN-based module that integrates semantic (text embeddings) and structural information. For each node pair, we compute structural features based on graph topology, such as common neighbors and the Adamic-Adar index. The model stacks GNN layers (e.g., GCN, GAT, and GraphSAGE) (Kipf & Welling, 2016; Veličković et al., 2017; Hamilton et al., 2017) to produce contextualized node embeddings. For any two nodes, their embeddings and structural features are concatenated and passed to a multilayer perceptron that outputs a link probability score. The model is trained with a binary cross-entropy loss, enhanced by a hard negative sampling strategy.

## 4 EVALUATION

In the evaluations, we aim to answer the following research questions:

- RQ1: How effective is LLM4CTI at knowledge graph construction from CTI texts?
- RQ2: How do key parameters (model selection , chunk size, temperature) impact effectiveness?
- RQ3: How effectively does LLM4CTI's GNN module perform link prediction?

### 4.1 EVALUATION SETUP

**Implementation**. Our system uses the open-source QWQ-32B model (Qwen, 2025), configured with a temperature of 0.2 and a maximum output of 32,768 tokens. Full implementation details are in Appendix G.

**Datasets**. Our evaluation relies on two curated datasets constructed from authoritative sources like Securitylist and MITRE ATT&CK, spanning 10 major threat categories:

- *Diverse CTI Dataset*: Used for knowledge graph construction (RQ1 and RQ2), this dataset comprises 55 articles. It features a manually annotated ground truth of 791 entities and 744 relationships, with an average information density of 14.38 entities and 13.53 relationships per article.
- *Top Security Entity Dataset*: Used for relationship prediction (RQ3), this dataset contains 302 articles focused on frequently co-occurring threat entities, forming a graph with a total of 11,494 nodes and 14,972 edges.

Full details on the dataset construction methodology are provided in Appendix F.

### 4.2 RQ1: KNOWLEDGE GRAPH GENERATION

In this RQ, we applied LLM4CTI to construct knowledge graphs from articles of the Diverse CTI Dataset.

**Evaluation Metrics** We assess the effectiveness of graph construction using the metrics of precision and recall. We define the metrics based on the classification of extracted triples. For precision, a True Positive (TP) is an extracted triple supported by the source text, while a False Positive (FP) is one that is not. Precision is then the fraction of extracted triples that are correct. For recall, we compare the output against the ground truth. A ground-truth triple successfully generated by the model is a True Positive (TP), while a ground-truth triple the model failed to extract is a False Negative (FN). Recall is then the fraction of ground-truth triples that are successfully recovered.

We used Gemini 2.5 Pro as a semantic matcher and verifier to assess whether each predicted triple semantically corresponds to a human-labeled ground truth triple, with explicit verification of textual support (prompts in Appendix H and I). The judge was validated against two human experts (computer science PhDs) on 600 triples. The inter-rater agreement was high (85.5% on precision, 87.3% on recall), confirming its reliability.

**Baseline Approaches**. We compare LLM4CTI with several state-of-the-art approaches, including both LLM-based and non-LLM methods. As a non-LLM baseline, we include **Extractor** (Satvat et al., 2021), which leverages BERT (Devlin et al., 2018) and BiLSTM (Hameed & Garcia-Zapirain, 2020) to extract attack behaviors. The primary LLM-based systems used for comparison are:

- **CTINexus** (Cheng et al., 2024): This approach processes the full text of a CTI article in a single pass. Consistent with its paper, we use GPT-4o as its backend model.
- **CTIKG** (Huang & Xiao, 2024): CTIKG first segments articles and uses a dual memory design to extract triples. We also use GPT-4o as its backend LLM, following its original implementation.
- **GraphRAG** (Microsoft, 2024): This approach also uses GPT-4o. To ensure a fair comparison, we provide GraphRAG with the same entity and relationship definitions used by LLM4CTI.

To ensure a fair comparison with a SOTA commercial model, we also benchmarked LLM4CTI with OpenAI's o4-mini. We selected OpenAI's o4-mini, as GPT-4o frequently failed due to its output token limit.

**Effectiveness Comparison**. Table 1 shows the performance of all approaches. The results highlight that the two LLM4CTI variants secure the top two positions in overall performance. LLM4CTI with o4mini achieves the highest F1 score of 85.13%, followed by the QWQ-based LLM4CTI at 81.58%, both outperforming the other baseline systems. When comparing against systems that also use OpenAI's models, LLM4CTI (o4mini) demonstrates a strong trade-off. While its precision (89.87%) is slightly lower than the category-leading GraphRAG (93.01%), its recall of 81.58% is significantly higher than all other LLM-based baselines, including GraphRAG (70.25%) and CTIKG (76.73%). Furthermore, when using a smaller, locally-deployed open-source model, LLM4CTI (QWQ 32B) remains highly effective, achieving an F1 score of 81.58% which is comparable to and even better than both CTINexus (75.27%) and CTIKG (78.98%). Although GraphRAG posts the highest precision, its recall of 70.25% is one of the lowest among the LLM-based methods, suggesting an imbalance in its extraction strategy. Finally, the non-LLM based Extractor performs significantly worse than all modern LLM-based approaches. This is largely because its methodology tends to extract long, complex subordinate clauses instead of discrete, well-defined entities. We also measure the impacts of article length in Appendix J.

**Cost Comparison**. The per-article cost for LLM4CTI with the QWQ model is approximately $0.0031 on a local GPU and $0.026 on RunPod.io (2025). In contrast, API-based methods are more expensive: LLM4CTI with o4-mini variant costs $0.19, while the GPT-4o-based baselines cost $0.07 (GraphRAG), $0.12 (CTIKG), and $0.026 (CTINexus), respectively.

**Node and Relationship Type Classification Accuracy.** To assess the accuracy of our schema-guided extraction, we analyzed the classification of entity and relationship types produced by the LLM4CTI (QWQ 32B). From the generated knowledge graphs, we randomly sampled up to 50 instances for each non-'other' entity and relationship type, creating a total evaluation set of 512 entity records and 670 relationship records. The results indicate that LLM4CTI achieves a high degree of accuracy in this task. Specifically, the entity type classification accuracy was 90.43%, and the relationship type classification accuracy reached 95.82%. This evaluation assesses type consistency, not extraction correctness (addressed in Table 1).

**Effectiveness of LLM4CTI's Schema**. Our evaluation results highlight this performance difference, particularly in recall. As shown in Table 1, when both systems use OpenAI backends,

Table 2: Micro-averaged performance of LLM4CTI with different LLMs

| Model | Precision | Recall | F1-Score |
|---|---|---|---|
| QWQ 32B | 86.21% | 80.29% | 83.15% |
| OpenAI o4-mini | 89.90% | 80.98% | 85.21% |
| GPT-4o | 57.20% | 44.29% | 49.92% |
| LLaMa4 Maverick | 67.30% | 56.88% | 61.65% |
| Qwen3 32B | 78.59% | 78.47% | 78.53% |
| Qwen3 30B-A3B | 79.94% | 68.30% | 73.66% |

LLM4CTI (o4mini) achieves a recall of 81.58%, which is over 15 points higher than CTINexus's 65.97%. While their precision is comparable (89.87% vs. 88.65%), this gap in recall gives our approach a decisive lead in F1 score (85.13% vs. 75.27%). The advantage in recall is also evident when using our locally-deployed model. While the precision of LLM4CTI (QWQ 32B) is 3.07% lower than that of CTINexus (85.58% vs. 88.65%), its recall is a substantial 12.69% higher (78.66% vs. 65.97%). We attribute this consistent advantage in recall to the enhanced specificity of our formal framework. The ground-truth data reveals that a significant portion of threat information (30.85% of entities; 25.40% of relationships) is unrepresented in existing STIX schema, underscoring the necessity of our expanded framework.

### 4.3 RQ2: Ablation Study on Model Selection and Parameters

In this section, we conduct studies to analyze how LLM4CTI's performance is affected by its backend LLMs and the chunk size.

**Impact of Different LLMs**. We first evaluated the performance of LLM4CTI using micro-averaged precision, recall, and F1 across different backend LLMs, with results summarized in Table 2. We compare the QWQ and OpenAI o4-mini models used in RQ1 against several other prominent models: GPT-4o, LLaMa4 Maverick (Meta AI Team, 2025), Qwen3 32B (Qwen Team, 2025), and Qwen3 30B-A3B. For the Qwen3 models, we enabled their built-in ability to automatically switch between direct outputs and reasoning outputs, whereas the QWQ model was trained to always produce reasoning outputs.

The results show that our originally selected models, OpenAI o4-mini and QWQ 32B, achieve the highest F1-scores (85.21% and 83.15%, respectively). The Qwen3-based models deliver respectable performance, but both GPT-4o and LLaMa4 Maverick perform significantly worse. A manual review of their outputs reveals that their low scores are primarily due to a failure to consistently adhere to the structured output format specified in the prompt. For example, GPT-4o often produced malformed or non-English text within its output: ""mother entity": ["None"] end tackle fencing ... amplify we'll of axis then context-context ...,". LLaMa4 Maverick outputs this python code instead of the required format: "print("#Final_Relationship_List_Start#") print(json.dumps(merged_relationships, indent=2)) print("#Final_Relationship_List_End#")".

**Impact of Temperature.** We also measure the impacts of temperatures for QWQ model (detailed in Appendix K), while OpenAI o4 mini doesn't provide the temperature adjustment. As detailed in Table 5 (Appendix K), performance is optimal at a low temperature of 0.2, but the F1-score plummets from 83.15% to 58.08% when raised to 0.6 as the model's output becomes chaotic. At higher temperatures, the model's output becomes increasingly chaotic and often fails to follow instructions, like this output: "Okay Okay nOkay¡think¿ Okay Okay Okay Okay ¡think¿ Okay Okay Okay. Okay."

**Impact of Chunk Size.** We find that chunk size also significantly impacts performance. As detailed in Table 6 (Appendix L), decreasing the chunk size of Qwen3 32B from 400 to 100 tokens boosts the F1-score from 78.53% to 82.30%. This gain is primarily driven by a substantial increase in recall (from 78.47% to 86.38%), likely because smaller chunks allow the model to focus on more granular details.

### 4.4 RQ3: SEMANTIC RELATIONSHIP PREDICTION

We evaluate the generated knowledge graph's utility via link prediction, where a GNN-based module predicts missing relationships.

**Evaluation Setup**. We use the *Top Security Entity Dataset*, partitioned by source article into a 70% training se t($G_{train}$ with 7,768 nodes/10,298 edges) and a 30%test set ($G_{test}$ with 3,726 nodes/4,674 edges). To handle semantic ambiguity (e.g., 'uses' vs. 'exploits', 'consists-of' vs. 'has', 'communicates-with' vs. 'downloads-from'), we simplify the task to predicting link existence (ignoring the specific relation type) and use the standard "Filtered" protocol for fair evaluation. Performance is measured using Mean Reciprocal Rank and Hits@K (K=1, 5, 10).

**Results and Analysis**. The link prediction results in Table 7 (Appendix M). When a new malware entity is added to the graph, the model can infer its connections to existing entities. For example, based on the performance shown in our evaluation, the system can predict a new malware's likely threat actor (malware $\rightarrow$ threat actor/intrusion), with the correct link appearing in the top 10 results 78.15% of the time (Hit@10). Similarly, it can suggest potential attack patterns (attack pattern $\rightarrow$ malware, Hit@10 of 62.50%) and, most effectively, its connection to other malware in a call chain (malware $\rightarrow$ malware), achieving an impressive 80.57% Hit@10 rate. This provides analysts with a ranked shortlist of immediate, actionable hypotheses for a new threat, significantly accelerating the initial triage and investigation process.

## 5 DISCUSSION

LLM4CTI uses detailed prompt to guide LLM. Some research (Thirunavukarasu et al., 2023) mentioned that fine-tuning improves LLM for a particular task. We attempted to fine-tune small LLM on the entity extraction task, but observed no significant improvement. Lobo et al. (2024) has shown that fine-tuning, especially on smaller models, can degrade the inherent LLM's Chain-of-Thought ability and increase hallucinations (Gekhman et al., 2024). As a result, we decided to rely on prompt engineering.

## 6 RELATED WORK

**CTI Articles.** Prior work on extracting intelligence from CTI articles has evolved from simple Indicator of Compromise extraction (Liao et al., 2016; Catakoglu et al., 2016), to statistical TTP identification using TF-IDF (Husari et al., 2017), and finally to ML-based TTP prediction (Das Purba et al., 2020; Sauerwein & Pfohl, 2022). However, these approaches are limited by their reliance on curated word lists and shallow semantic understanding. Our approach, in contrast, leverages large language models to capture richer contextual semantics from CTI texts.

**Knowledge Graph.** Knowledge graphs are widely applied in cybersecurity, with prior work focusing on creating malware ontologies (Rastogi et al., 2020), recording threat behaviors (Piplai et al., 2020), detecting malicious nodes (Najafi et al., 2019), and structuring CVE data from sources like the NVD (Qin & Chow, 2019; Høst et al., 2023). More recently, researchers have begun using LLMs for this task (Yao et al., 2023). Notably, both AttackG+ (Hu et al., 2024) and CTIKG (Huang & Xiao, 2024) also leverage LLMs to construct knowledge graphs from CTI reports, with the latter being a key open-source baseline.

## 7 CONCLUSION

We propose LLM4CTI, a novel LLM-based method for generating cyber threat knowledge graphs from unstructured CTI articles. LLM4CTI segments articles into chunks and uses a chunk-wise pipeline with a dual-context input to reduce hallucinations and capture long-range relationships. A formal modeling framework helps filter irrelevant entities and focus on threat-relevant relations. Evaluations on real-world articles show that LLM4CTI outperforms SOTA methods in knowledge graph construction can support downstream tasks like cross-article semantic relationship prediction.

ETHICS STATEMENT

The Cyber Threat Intelligence articles used for our dataset were collected from publicly accessible sources, and we have ensured that the data contains no Personally Identifiable Information or other sensitive private data.

REPRODUCIBILITY STATEMENT

The full source code for the LLM4CTI framework with all data processing scripts, the knowledge graph construction pipeline, the GNN module, and evaluation scripts is available at our anonymous project repository: `https://github.com/hiddenauthor/LLM4CTI`. This repository also contains the datasets, corresponding ground truth and evaluation prompt used in our experiments. For convenience, abridged versions of the core framework prompts are provided in Appendix C and D. All hyperparameters are detailed in the Evaluation section.

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

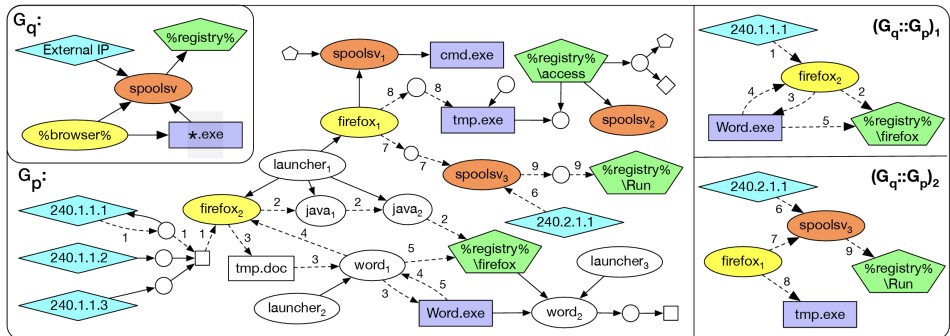

Figure 4: Aligning CTI knowledge graph with provenance graph built from system auditing events Milajerdi et al. (2019)

Petar Veličković, Guillem Cucurull, Arantxa Casanova, Adriana Romero, Pietro Lio, and Yoshua Bengio. Graph attention networks. *arXiv preprint arXiv:1710.10903*, 2017.

Thomas D Wagner, Khaled Mahbub, Esther Palomar, and Ali E Abdallah. Cyber threat intelligence sharing: Survey and research directions. *Computers & Security*, 87:101589, 2019.

Y. Wang, Y. Zhang, Y. Li, and X. Liu. A bibliometric review of large language models research from 2017 to 2023. *arXiv preprint arXiv:2304.02020*, 2023.

Liang Yao, Jiazhen Peng, Chengsheng Mao, and Yuan Luo. Exploring large language models for knowledge graph completion. *arXiv preprint arXiv:2308.13916*, 2023.

Yongheng Zhang, Tingwen Du, Yunshan Ma, Xiang Wang, Yi Xie, Guozheng Yang, Yuliang Lu, and Ee-Chien Chang. Attackg+: Boosting attack knowledge graph construction with large language models. *arXiv preprint arXiv:2405.04753*, 2024.

Sihang Zhao, Youliang Yuan, Xiaoying Tang, and Pinjia He. Difficult task yes but simple task no: Unveiling the laziness in multimodal llms. *arXiv preprint arXiv:2410.11437*, 2024.

# Appendices

## A    ALIGNING CTI KNOWLEDGE GRAPH WITH SYSTEM AUDITING EVENTS

Figure 4 shows an example query graph ($G_q$) generated from a CTI knowledge graph, and a simplified provenance graph ($G_p$) built from system auditing events Milajerdi et al. (2019). The right side of the figure shows two sample graph alignments ($G_q :: G_p$). Different node shapes are used to distinguish node types, while nodes with the same color denote possible alignments. The two alignments illustrate attack behaviors in $G_p$ that match the patterns specified in $G_q$, which is derived from the knowledge graph. This result highlights both the importance and effectiveness of constructing a precise knowledge graph from CTI articles that capture a wide range of attack scenarios.

## B    FORMAL REPRESENTATION OF KNOWLEDGE GRAPH

We formally define the components of a knowledge graph below.

**Entity (Node).** An `entity` refers to any unit of information that can be independently identified, described, and analyzed in the context of cyber threat intelligence. While conventional approaches often emphasize IOCs such as IP addresses and domain names, our definition covers a broader range of threat-relevant components. These include named or unnamed attacker tools, function names,

malware modules, specific attack techniques, and even semantically meaningful but unnamed arti-facts (e.g., a downloaded *RAR file*).

An `entity` must satisfy the following properties:

- **Independence:** The entity functions as a standalone unit of information that can be extracted or analyzed in isolation.
- **Relevance:** The entity is meaningfully connected to other entities and contributes to the recon-struction of attack chains or threat narratives.
- **Diversity:** Valid entities span across various data types, including files, scripts, malware modules, registry keys, attack methods, and observable network traces.
- **Contextual Significance:** Even without an explicit name, an entity is considered valid if it plays a meaningful role within its contextual setting.

An `entity` is defined to possess the following structured attributes:

- **name**: The canonical name of the entity.
- **type**: A label selected from a predefined taxonomy of entity types.
- **alias**: A list of alternative names or surface mentions for the entity found within the source text.
- **parent entity**: The name of the parent entity if the entity is a subcomponent of another (e.g., a malware module or function).

Our entity schema extends the Structured Threat Information eXpression (STIX) 2.1 standard TC (2017), a structured language for representing Cyber Threat Intelligence (CTI). To systematically identify the limitations of the standard, we first used the baseline STIX 2.1 schema to classify entities from 782 CTI articles with OpenAI's o4-mini model. We then manually reviewed all entities the model failed to classify and, based on this analysis, created a new set of entity types to address the observed gaps.

This process resulted in our final extended schema. Each category is defined with a definition, differentiation notes with other categories, and examples to guide LLM.

**Relationship (Edge).** A **relationship** defines the semantic connection between two entities, cap-tured by a surface-level phrase (`rel`) and a mandatory, normalized category (`rel_type`). The process for populating these fields distinguishes between explicit and implicit connections found in the source text.

For **explicit relationships**, where a clear connecting verb or phrase exists (e.g., "send a slack mes-sage to"), this text is first extracted as the value for `rel`. This surface phrase is then classified into its corresponding normalized category from our taxonomy to determine the `rel_type`.

For **implicit or long-distance relationships**, where no such connecting phrase is present, the pro-cess is inverted. The LLM's primary task is to infer the semantic connection and assign the most appropriate category from our taxonomy as the `rel_type`. In these cases, to ensure the triple is complete, the `rel` field is then populated with the value of the inferred `rel_type` (e.g., *indicates*).

**Triple.** A `triple` is the atomic unit of the knowledge graph and represents a fact extracted from the text. A `triple` consists of:

- **sub**: the subject entity,
- **rel**: the surface-level relationship phrase,
- **obj**: the object entity.

## C    PROMPT FOR CHUNK-LEVEL KNOWLEDGE EXTRACTION

**User**: You are an NLP model for extracting cybersecurity threat knowledge graphs from text by identifying entities and relationships.

**What is an entity:**    In cyber threat intelligence, an entity is any identifiable unit of information relevant to threat activity. It includes not only IOCs like IPs or domains, but also broader

objects such as scripts or generic items like "RAR file," even without fixed names. The main characteristics of entities are:

1. **Independence:** Each entity exists as an independent unit of information and can be extracted and analyzed separately.

2. **Relevance:** There may be inherent connections between entities, and by associating this information, a complete attack chain or threat portrait can be constructed.

3. **Diversity:** Entities can cover various forms of information such as files, scripts, configuration files, registry entries, network traffic data, log records, etc.

4. **Contextual significance:** Even if an entity does not have a specific name, as long as it has contextual significance and analytical value in threat intelligence analysis, it is still a valid entity.

**Stage 1 – Entity Extraction and Classification**

- **Stage 1.1:** Fully scan the current chunk to identify all entities related to cybersecurity threats, including explicitly named indicators, implicit threat components, and contextually significant objects even without specific names. Only extract entities from named entities that directly appear in the current chunk. If it is discovered that an entity in the current chunk has an relationship with a outside-chunk entity , add it to the entity list and record that relationship later.

- **Stage 1.2 – Type Assignment:** Assign one predefined category to each entity. Use threat-actor/intrusion-set formatted names for unnamed attackers or attacks. Apply another category only when no predefined type matches.

- **Stage 1.3 – Alias & Lineage Handling:** Record all aliases or alternative names for entities. Identify evolutionary relationships for the "parent entity" field where applicable.

- **Stage 1.4 – Recheck:** Before finalizing entity extraction, pause and critically reflect on whether any relevant entities may have been overlooked, including both explicitly named and contextually referenced but unnamed entities.

**Predefined entity types:** ¡Text from Entity Schema Document¿

**Stage 2 – Relationship Extraction and Classification**

- **Stage 2.1:** Extract direct and indirect relationships (`rel`) among entities in the current chunk and classify them into predefined relationship types (`rel_type`). The `rel` should be taken directly from the chunk text and simplified only when necessary. The `rel_type` must be selected from the predefined types in the Supplement; if none applies, use `other`. In most cases, `rel` should differ from `rel_type`. Only when the relationship is not clearly expressed in the text may a `rel_type` be reused as `rel`, based on context.

- **Stage 2.2:** If an entity in the current chunk has a relationship with an entity not directly mentioned in the chunk but in the full text (e.g., through indefinite pronouns, chapter references, or context hints), add that outside-chunk entity to the entity list and extract the corresponding relationship. A common example is when the current chunk has a list of multiple URLs/hashes/filenames/domains without any specific relationship, but you should think if the list of entities has a relationship with the topic threat entity. And the listing means those "entities" are related to the topic threat entity with 'indicates' or 'characterizes' relationship.

- **Stage 2.3 – Recheck:** Ensure each entity has at least one relationship. If not, infer one from context or link it to a main entity outside the chunk and add both to the output.

**Supplement: Predefined `rel_type`** ¡Text from Relationship Schema Document¿

**Stage 3 – Output Format**
**Normalization of Obfuscated URLs, IPs, and Emails:**

Obfuscated URLs, IP addresses, and email addresses must be converted to their original format. Specifically, replace `[.]` with `.` in URLs and IPs, and replace # with @ as well as `[.]` with `.` in emails.

**Part 1: Entity List**
Each entity node must include the following attributes, with all values represented as strings or string arrays: *name*, *type*, *alias*, and *parent entity*.

**Part 2: Entity Relationships**

- Extract relationship descriptions between entities from the text and output them as a JSON array of objects, with keys sub, rel, rel_type, and obj.

- Each relationship object must follow this format:

    { "sub": "*<Source Entity>*", "rel": "*<Relationship Text>*", "rel_type": ["*<Relationship Type Category>*"], "obj": "*<Target Entity>*" }

Below is an example: *<Examples>*. Now here is the full text: *<Full text>* Here is the current chunk's text: *<Chunk Text>*

## D  PROMPT FOR ARTICLE-LEVEL KNOWLEDGE GRAPH CONSTRUCTION

**User**: "You are working on merging results from a distributed knowledge graph construction task for cybersecurity threat intelligence. A single article was previously split into multiple chunks. Each chunk has gone through entity and relationship extraction. Now, you need to merge the processing results of these chunks to form a complete article-level knowledge graph."

**Merging Rules**

1. **Entity Merging:** Consider two entities the same if their names are identical or semantically equivalent (including case differences).
   **Alias Name Handling:** Use the simplest surface form as the canonical name, and record other variants in the alias list. Add a variant relationship triple. Finally, merge alias lists across chunks, removing duplicates.

2. **Parent Entity Handling:** For any entity with a non-empty parent entity field, generate a structural relationship triple. All such parent-child triples must be included.

3. **Relationship Merging:** Merge relationships only if the subject, object, and relationship description are all identical.
   **Merging Strategy:** Merge different relationship types into a set and keep the original text.

4. **Implicit Link Completion:** If two entities are implicitly related across chunks through shared parent entities or contextual clues, add an inferred relationship between them.

5. **Must-Following Rules:** Do not discard anything, just merge. Keep aliases, parent entities, and all original details while removing duplicates.

**Output Requirements**
Maintain the same JSON structure as the original.

**Actual Input**
Below are the entities and relationships I extracted from multiple chunks: *<Actual entity list and relationship list from various chunks>*.

## E    ENTITY AND TRIPLE DEDUPLICATION PROCESS

The deduplication process consists of two main phases: entity deduplication and triple deduplication, detailed as follows:

- **Entity Deduplication Rules**: Two entities are treated as equivalent if their names are identical or semantically equivalent, including case-insensitive variants.
- **Alias Name Handling**: The simplest surface form is selected as the canonical name. All alternative names are merged into the *alias* list, and duplicates are removed. For example, if "APT28" and "APT28 (Fancy Bear)" appear in different chunks, the merged entity is named "APT28", and "Fancy Bear" is recorded as an alias.
- **Parent Entity Handling**: If an entity is a component or subpart of another, this structural dependency is recorded in the *parent entity* field. An additional triple (e.g., *variant-of*) is added to explicitly represent this connection in the knowledge graph.
- **Triple Deduplication Rules**: Two triples are merged only if their subject (*sub*), object (*obj*), and surface-level relation phrase (*rel*) are exactly the same after entity unification. The *rel_type* is merged into a set of distinct values.
- **Implicit Link Completion**: If two entities appear in different chunks but are implicitly related, such as through a shared parent entity or contextual clues, LLM4CTI explicitly adds a new triple to connect them. This ensures the resulting knowledge graph remains complete and well connected.
- **Merging Integrity Constraints**: No entity or triple is discarded unless it is an exact duplicate. If complementary attributes exist, such as additional aliases or parent entity links, all such information is preserved in the final merged result.

## F    DATASET CONSTRUCTION AND DETAILS

To construct our evaluation datasets, we first gathered a corpus of $88,803$ CTI articles from $67$ websites and $672$ articles from the ATT&CK knowledge base. Due to the significant manual effort required to create ground-truth knowledge graphs from this large corpus, we curated two smaller, focused datasets as described in the main body.

- *Diverse CTI Dataset*: A set of 55 articles was used to evaluate knowledge graph construction (RQ1 and RQ2). To establish the ground truth, we first used Gemini 2.5 Pro to generate a draft knowledge graph for each article, which was then manually reviewed and corrected by verifying each fact, removing errors, and adding omissions. The articles have an average of 1,057 tokens (min: 165, max: 2,504), and their ground-truth graphs contain an average of 14.4 entities (min: 2, max: 49) and 13.5 relationships (min: 1, max: 47).
- *Top Security Entity Dataset*: A set of 302 articles was used for relationship prediction (RQ3), focused on "frequently co-occurring" threat entities. We define this term as the top five security entity pairs that most often appear together within the same article in our corpus (e.g., the *SpyEye, Zeus* pair). The knowledge graph for this dataset was constructed using the LLM4CTI (o4mini) model and, after merging, contains 11,494 entities and 14,972 relationships.

The detailed curation process for the *Top Security Entity Dataset* was as follows:

We applied an LLM-based query to extract 18,220 articles from our full corpus that describe threat behaviors associated with security entities. Among these, we identified the top five security entity pairs that most frequently appear within the same article. We then collected all articles that contain at least one of these top security entity pairs, resulting in the final curated dataset of 302 articles.

The tables below provide further details on the curated datasets.

## G    IMPLEMENTATION DETAILS

Our system is implemented with Vllm (Kwon et al., 2023), NetworkX (developers, 2023), spaCy (Explosion AI, 2024), and TensorBlock (TensorBlock, 2025). The open-source QWQ-32B (Qwen, 2025) model is self-hosted via Vllm on a server with dual Nvidia RTX 6000 Ada GPUs, while commercial OpenAI and Gemini models are accessed through the TensorBlock provider.

Table 3: Threat categories in the Diverse CTI Dataset

| Threat Category | Count | Percentage (%) |
|---|---|---|
| APT | 6 | 10.9 |
| Adware | 5 | 9.1 |
| Botnet | 5 | 9.1 |
| Denial-of-Service Attack | 3 | 5.5 |
| Ransomware | 7 | 12.7 |
| Spam/Phishing | 9 | 16.4 |
| Supply Chain Attack | 5 | 9.1 |
| Trojan horse | 3 | 5.5 |
| Virus/Worm | 7 | 12.7 |
| Vulnerability/exploit | 5 | 9.1 |
| Total | 55 | 100.0 |

Table 4: Top security entity dataset

| Threat Entity(Pair) | Frequency |
|---|---|
| Zeus | 93 |
| SpyEye | 55 |
| WannaCry | 49 |
| Blackhole | 45 |
| EternalBlue | 35 |
| SpyEye, Zeus | 35 |
| EternalBlue, WannaCry | 33 |
| Locky, Necurs | 16 |
| Blackhole, Zeus | 12 |
| Koobface, Zeus | 11 |

## H  PROMPT FOR PRECISION EVALUATION

**User**: You are a professional AI assistant specializing in evaluating the Precision of Knowledge Graph (KG) entity relationships extracted from text. Your task is to receive a "Source Text", a list of "Predicted Values" relationships, and a "Ground Truth" relationship list for reference only. You need to evaluate each relationship in the "Predicted Values" list and output the detailed evaluation results in the specified JSON format.

**Evaluation Attitude: Strive to Confirm, Cautious to Falsify**
Your core evaluation principle is "Strive to Confirm, Cautious to Falsify". Consider yourself the "defense attorney" for the prediction model, not the "prosecutor". Your primary task is to do everything possible to find a chain of evidence in the "Source Text" and "Advanced Reasoning Rules" that can prove the "Predicted Values" are correct.

> **Default Trust:** Unless a predicted relationship has a clear, irreconcilable, and direct contradiction with the "Source Text", you should prioritize assuming it is correct (TP).
>
> **Burden of Proof:** Judging a relationship as a False Positive (FP) requires strong, direct counter-evidence. Merely "not being directly mentioned in the source text" or "requiring multi-step reasoning" are never sufficient reasons to classify it as an FP.
>
> **Acknowledge Reasoning:** Place high value on the abstraction, generalization, and reasoning capabilities demonstrated by the model. Your job is to verify whether this reasoning is logical and well-founded, not just to perform literal string matching.

**Workflow**

1. **Iterate Through Predictions:** Start with the first relationship in the "Predicted Values" list, `predict_relationship[i]`, and evaluate them one by one.

2. **MANDATORY Subject Attribution Analysis:** This is the first analysis that must be performed before any matching or verification. Analyze the subject of the current predicted relationship and quickly scan the entire text to answer a core question: "Are there any entities in the source text that act as a proxy, tool, component, or actor for this subject?" Based on this analysis, conceptually establish a temporary list of equivalent entities.

3. **Prioritize Matching with Ground Truth:** Compare the current `predict_relationship[i]` with the entire "Ground Truth" list. During the comparison, you must use the equivalence list established in Step 2 and apply all "Advanced Reasoning Rules". If a match is found, directly judge it as TP and proceed to the next prediction.

4. **Final Verification with Source Text:** If no match is found in the "Ground Truth" list, you must directly verify `predict_relationship[i]` against the "Source Text", again using the equivalence list and all "Advanced Reasoning Rules". If supported, judge as TP. If not supported or contradicted, judge as FP.

**Advanced Reasoning Rules**
These rules are the basis for your complex judgments and can be used both when matching "Ground Truth" and when verifying against the "Source Text".

- **Rule 1: Chain Deduction Equivalence Rule.** If a relationship A -¿ C can be logically deduced from a chain (e.g., A -¿ B and B -¿ C), it is equivalent. *<Example Area>*

- **Rule 2: General-Specific Equivalence Rule.** If one entity is a general or specific description of another (e.g., "Google Play" vs. "apps on Google Play"), they are considered equivalent. *<Example Area>*

- **Rule 3: Action-Technique Equivalence Rule.** A specific action (e.g., "injects code into DLL") is equivalent to its corresponding named technique (e.g., "employs DLL Hollowing"). *<Example Area>*

- **Rule 4: Event-Element Complementarity Rule.** Relationships describing different elements of the same event (e.g., action+object and action+destination) are considered equivalent. *<Example Area>*

- **Rule 5: Relation Semantic Equivalence Rule.** Relationships with equivalent subjects/objects are equivalent if their relation verbs express similar intent (e.g., "incorporated for disguise" vs. "uses"). *<Example Area>*

- **Rule 6: Role Inversion and Functional-Structural Equivalence Rule.** A functional relationship (e.g., A uses B) can be equivalent to an inverted structural relationship (e.g., B is-part-of A). *<Example Area>*

- **Rule 7: Exclusion and Rejection Rule.** Reject malformed extractions where the subject or object is not a clear named entity (e.g., is a full sentence, clause, or pronoun). *<Example Area>*

- **Rule 8: Placeholder Entity Resolution Rule.** Resolve placeholder entities like 'Attacker(using: X)' by adopting the context ("the attacker associated with X") and looking for fused evidence in the source text. *<Example Area>*

- **Rule 9: Canonical Relation Validation Rule.** If a relation verb is a canonical name from a predefined list, validate the relationship based on the verb's definition, not its literal presence in the text. *<Example Area>*

- **Rule 10: Relationship Hierarchy Inclusion Rule.** When predicted and ground truth relations have a hierarchical relationship (e.g., 'downloads' is a subclass of 'communicates-with'), judgments should be inclusive based on whether the prediction is more specific or more general. *<Example Area>*

- **Rule 11: Entity Attribute Validation Rule.** Validate alias or variant relationships by checking the 'alias' or 'mother_entity' attributes in the Ground Truth entity list, recognizing the equivalence between representing facts as edges versus attributes. *<Example Area>*

- **Rule 12: Entity Alias Equivalence Rule.** If A is an alias of B, they are fully interchangeable in any relationship. A triplet '(A, rel, C)' is equivalent to '(B, rel, C)'. *<Example Area>*

- **Rule 13: Entity Hierarchy Inheritance/Induction Rule.** If A's 'mother_entity' is B, behaviors can be inherited downwards (B to A) or induced upwards (A to B). '(A, rel, C)' is equivalent to '(B, rel, C)'.

**<Multiple Examples>** (Specific examples are available at our project repository: `https://github.com/hiddenauthor/LLM4CTI`.)

# I  PROMPT FOR RECALL EVALUATION

**User**: You are a professional AI assistant responsible for evaluating the accuracy of Knowledge Graph (KG) entity relationships extracted from text. Your task is to receive a "Source Text," a list of "ground truth" relationships, and a list of "predict values" relationships. You will then compare each item in the "ground truth" list to determine if it exists in the "predict values" list.

**Evaluation Attitude: Dig Deep for Connections, Strive to Match**
Your core evaluation principle is to "Dig Deep for Connections, Strive to Match." You must assume that for every "Ground Truth" relationship, there is a very high probability of a semantically equivalent counterpart in the "Predicted Values" list. Your task is to use all your reasoning abilities to find this connection.

> **Assume a Match Exists:** Before exhausting all "Advanced Reasoning Rules" for inference and matching, you must never hastily judge a "Ground Truth" relationship as "FN" (False Negative).
>
> **Burden of Proof:** Judging a relationship as "FN" is the last resort. It implies that you have confirmed that the prediction model completely missed this fact.
>
> **Acknowledge Indirect Evidence:** A fact can be stated directly or implied through multiple indirect relational chains. Your job is to maximize the identification of facts that the prediction model has successfully "recalled."

**Core Evaluation Rules**

1. **Comparison Scope:** Only compare the core triplet ('sub', 'rel', 'obj'). Auxiliary data like 'alias' and 'mother_entity' can be used as supplementary evidence.

2. **Equivalence Rules:** Consider relationships equivalent based on core entity matching, active/passive voice, semantic equivalence of 'rel', and chain deductions (multi-hop relations).

3. **Exclusion Rules:** Ignore malformed extraction results where 'sub' or 'obj' is not a clear, specific entity.

4. **Use the Source Text as the Ultimate Authority:** Use the "Source Text" as the final arbiter for resolving all ambiguities, performing chain deductions, and making complex judgments.

5. **Advanced Rules:** Refer to the advanced rules to identify all possible equivalent relationships.

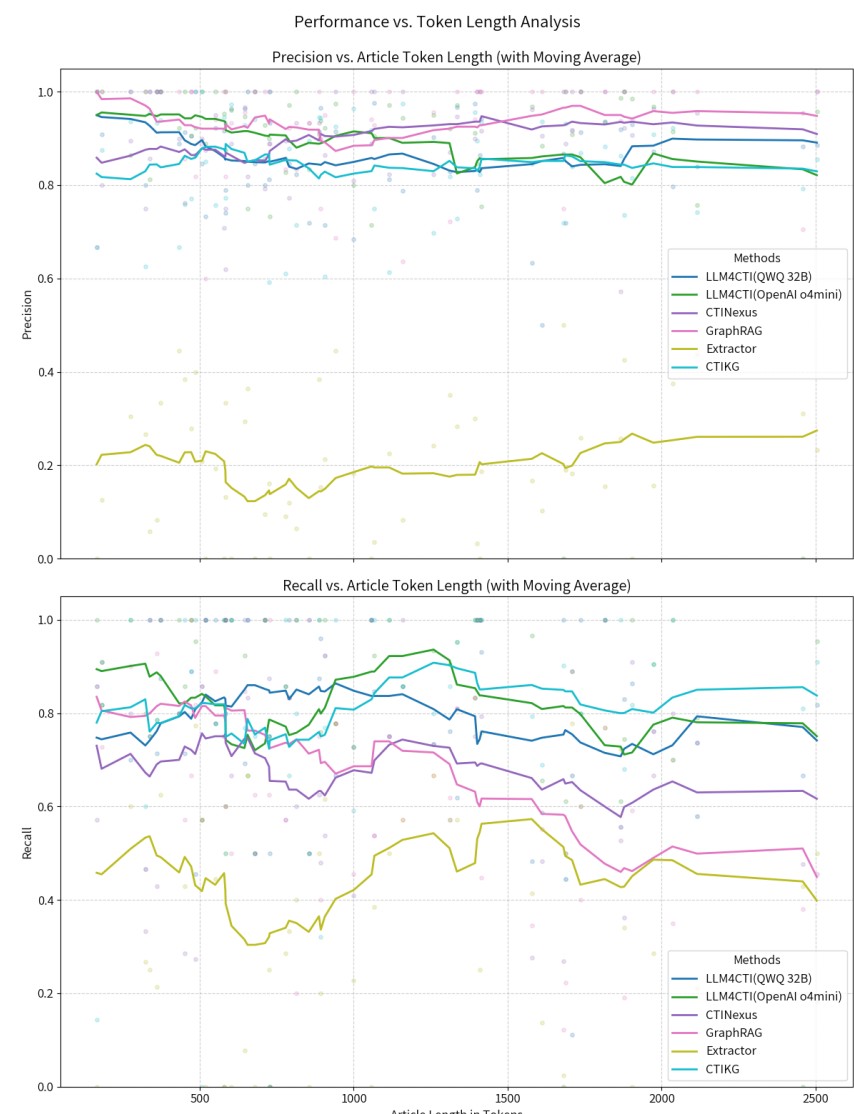

Figure 5: Impacts of article lengths

---

**Advanced Rules Explained**

The same advanced reasoning rules from the Precision prompt (Chain Deduction, General-Specific, etc.) should be applied here to find a match for each ground truth relationship.
  <**Multiple Examples**> (Specific examples are available at our project repository: `https://github.com/hiddenauthor/LLM4CTI`.)

---

## J  IMPACTS OF DIFFERENT ARTICLE LENGTHS

Figure 5 further illustrates the effectiveness of each approach across varying article lengths, with trend lines smoothed by a moving average. While the precision of most LLM-based approaches remains relatively stable across different document lengths, their recall shows a clear tendency to decline as articles get longer. This suggests a common challenge for LLMs in maintaining comprehensive contextual awareness over extended sequences of text.

This degradation in recall is particularly pronounced for methods that process the full text in a single pass. For example, GraphRAG's recall drops sharply from 66.12% on articles over 500 tokens to just 45.99% on those exceeding 1,500 tokens. CTINexus also sees its recall fall from 68.16% to 57.75% for the same length categories. In contrast, approaches that utilize a chunking strategy demonstrate greater resilience. LLM4CTI and CTIKG, which segment articles before processing, maintain more consistent recall. LLM4CTI, which combines its chunk-based processing with a global context view, mitigating the negative impact of increased article length while maintaining the ability to capture long distance entity relationship.

## K  ABLATION STUDY ON TEMPERATURE

Table 5: LLM4CTI (QWQ 32B) with different temperatures

| Temperature | Precision | Recall | F1-Score |
| --- | --- | --- | --- |
| 0.2 | 86.21% | 80.29% | 83.15% |
| 0.6 | 68.53% | 50.40% | 58.08% |
| 1.0 | 62.81% | 38.70% | 47.89% |

## L  ABLATION STUDY ON CHUNK SIZE

Table 6: LLM4CTI (Qwen3 32B) with different chunk sizes

| Chunk Size | Precision | Recall | F1-Score |
| --- | --- | --- | --- |
| 400 | 78.59% | 78.47% | 78.53% |
| 200 | 74.86% | 83.68% | 79.02% |
| 100 | 78.59% | 86.38% | 82.30% |

## M  EVALUATION RESULTS OF SEMANTIC LINK PREDICTION

Table 7: Results of link prediction

| Entity Pair Type | Hit@1 | Hit@5 | Hit@10 |
| --- | --- | --- | --- |
| malware → malware | 9.14% | 64.00% | 80.57% |
| malware → threat actor/intrusion | 3.36% | 57.98% | 78.15% |
| infrastructure → malware | 6.90% | 49.43% | 63.22% |
| general software → malware | 5.33% | 34.67% | 45.33% |
| attack pattern → malware | 2.78% | 54.17% | 62.50% |
| infrastructure → threat actor/intrusion | 11.11% | 50.79% | 71.43% |
| attack pattern → threat actor/intrusion | 3.57% | 42.86% | 71.43% |
| threat actor/intrusion → threat actor/intrusion | 2.33% | 60.47% | 74.42% |
| attack pattern → infrastructure | 3.33% | 53.33% | 60.00% |
| attack pattern → attack pattern | 7.41% | 74.07% | 81.48% |

## N    LLM Usage Statement

To evaluate at scale, we used Gemini 2.5 Pro as an automated judge to compare knowledge graphs against a manual ground truth. This approach was validated by achieving over 85% inter-rater agreement between the LLM's assessments and those of human experts.

