# OpenReview forum: "LLM4CTI: Uncovering Security Entities and Their Interactions from Unstructured Cyber Threat Intelligence"
_ICLR.cc/2026/Conference — Submitted to ICLR 2026_

### Official Review · Reviewer_zCG8 · 2025-10-28

**Soundness:** 2
**Presentation:** 2
**Contribution:** 1
**Rating:** 2
**Confidence:** 5

**Summary:**

This paper presents LLM4CTI, an LLM-based framework for extracting structured cyber threat intelligence (CTI) knowledge from long, unstructured CTI articles and transforming it into a knowledge graph. The method introduces a dual-context, chunk-wise processing pipeline that gives an LLM both local and global context to mitigate issues like “lost in the middle” and hallucination. It is guided by a formal threat-centric schema inspired by STIX 2.1 and integrates a Graph Neural Network (GNN) for downstream relationship prediction among security entities. Experiments on large CTI corpora show that LLM4CTI achieves state-of-the-art performance.

**Strengths:**

- **Originality**: Introduces a dual-context, chunk-wise reasoning framework that balances local accuracy and global context in long CTI articles—a novel application of LLMs to cyber threat intelligence.
- **Clarity**: Features a clean structure with step-by-step explanations, clear phase definitions, and illustrative diagrams for ease of understanding.
- **Significance**: Delivers open-source tooling that enables scalable, structured extraction and prediction of threat intelligence from unstructured text, outperforming commercial systems using open-source models.

**Weaknesses:**

- **Novelty**: While the paper proposes combining local and global context in its pipeline, this concept is not new; previous work has already addressed this challenge in both training and inference phases (e.g., via adaptive chunking) [1, 2]. Moreover, the fixed segmentation of documents into 400-token chunks appears arbitrary and lacks methodological justification.
- **Unfair Comparison**: In comparing its approach to the three baselines (CTINexus, CTIKG, and GraphRAG), the paper uses GPT-4o as the baseline model. Notably, Table 2 shows that LLM4CTI achieves only ~50% F1 when using GPT-4o, which raises questions about the efficacy and generalization of the proposed method.
- **Lack of Analysis**: The presentation of results lacks depth as there is minimal discussion of when or why the approach succeeds, fails, or underperforms. For instance, the link-prediction outcomes reported in Table 7 are hard to interpret without a clearly defined baseline or detailed breakdown of error modes.


[1] An, Shengnan, et al. "Make your llm fully utilize the context." Advances in Neural Information Processing Systems 37 (2024): 62160-62188.


[2] Duarte, André V., et al. "Lumberchunker: Long-form narrative document segmentation." arXiv preprint arXiv:2406.17526 (2024).

**Questions:**

- While the authors briefly discuss why GPT-4o performs so poorly with the proposed system, the other three baselines still perform much better with GPT-4o. Can the authors provide the corresponding baseline results using o4-mini or QWQ 32B, or explain why those baselines do not suffer from the same performance degradation as LLM4CTI?
- For RQ3, a proper baseline is needed to gauge the difficulty of the task and the size of the improvement achieved. Please include such a baseline in the evaluation.

---

> ### Author Response · Authors · 2025-11-19
>
> Thank you for your review and comments.
>
> For the novelty: We acknowledge that chunking techniques have been used in prior work to process long texts. However, our main contribution is not the chunking method itself, but rather the systematic framework we have developed. This framework constrains and guides the behavior of LLM on CTI text through precise task definitions, a formal ontology, and a multi-stage pipeline. The core value of our work is demonstrating that, with this carefully designed framework, even locally deployed open-source models such as QWQ 32B can achieve and even surpass the state-of-the-art performance previously attainable only by large, commercial closed-source models like GPT-4o.
>
> On the 400-token chunk size: The 400-token setting was not an arbitrary choice. We already conducted an ablation study on chunk size in Appendix L and Table 6 (comparing 400, 200, and 100 tokens). The results show that smaller chunks (like 100 tokens) can further improve the F1-score (from 78.53% to 82.30%). However, this is a trade-off: as the chunk size decreases, the processing time increases linearly. Furthermore, with the 100-token setting, the prompt used in the final merge step hits the 32k token limit.
>
> On the performance of GPT-4o: GPT-4o's poor performance within our framework is not because our method is ineffective, but due to limitations of the model itself. Our method is designed to achieve high recall, which generates very long and detailed reasoning chains. This causes GPT-4o to frequently hit its output token limit (16k), resulting in truncated output. In contrast, the prompts for CTINexus and GraphRAG are much simpler, so they do not hit this token limit, but they do so at the cost of much lower recall.

---

> ### Author Response · Authors · 2025-11-23
>
> To verify whether LLM post-training can replace our prompt design, we trained Qwen3-4B-Instruct-2507 with SFT and RL under a one-pass setting, removing both chunking and dual-context processing. The RL reward combined two components: (1) a format reward, encouraging the model to generate step-by-step reasoning traces, and (2) a result reward, defined as the correctness of multiple-choice questions that Gemini 2.5 Pro automatically generated from 800 CTI articles and their corresponding auto-constructed knowledge graphs. Evaluation followed the same protocol as in our main paper, using the 55 manually annotated CTI articles as ground truth and Gemini 2.5 Pro as the semantic judge. We observed that after training, the 4B model developed step-by-step reasoning behavior, but its language-generation stability degraded, and it sometimes failed to complete extraction within the 32k-token context window. Consequently, only 19 articles were successfully processed by all three settings within 32k tokens, and we report results on this overlap. On these 19 articles, the base Qwen3-4B achieved Precision 0.776, Recall 0.582, F1 0.614; Qwen3-4B with SFT achieved Precision 0.718, Recall 0.660, F1 0.640; and Qwen3-4B with SFT + RL achieved Precision 0.698, Recall 0.682, F1 0.632. RL increased recall but reduced precision, yielding no net F1 gain over SFT.
>
> Importantly, under our hardware environment (two RTX 6000 Ada GPUs) there is a practical trade-off between model size and context capacity: we can deploy larger models (e.g., a 72B model) for inference but cannot maintain a 32k-token context window when doing so, whereas full-scale training is limited to models of roughly 4B with practical context windows of about 4k tokens for both input and output. This resource-driven constraint further motivates our prompt-driven dual-context pipeline, which enables smaller, locally deployable models to achieve and even surpass the performance of commercial LLMs without requiring expensive hardware.

---

### Official Review · Reviewer_D5Jt · 2025-10-31

**Soundness:** 1
**Presentation:** 2
**Contribution:** 1
**Rating:** 2
**Confidence:** 5

**Summary:**

This paper proposes LLM4CTI to extract CTI evidence from long reports via dual-context, chunk-wise prompts (local chunk + full article) and merges results into a STIX-inspired knowledge graph. It then trains a GNN to predict latent links (e.g., malware–actor). Experiments show state-of-the-art F1 with notably higher recall than baselines and strong Hits@10 for link prediction. The problem is important to both AI and cybersecurity.

**Strengths:**

The methodology design is clear. The paper relies on a KG structure construction and conduct its design in a sequential way: extracting entities --> parse relations from unstructured texts.

**Weaknesses:**

1. There is a significant concern of novelty. Overall, this paper aims to construct KGs from articles, however, existing works such as ThreatKG [1] already did the same thing. How could this work claim its novelty and significance?

2. CTI from articles are limited in its scope. Mostly, CTI is collected from public threat feeds, daily reports, system logs, or vulnerability databases. Targeting only on articles are quite constrained on the impact and comprehensiveness.

3. The design heavily relies on AI-generated annotation or identification. The author also claims that using Gemini-2.5 achieve 85% agreement with human annotators, which implies a significant noise level (15%) in the produced KG.

4. The evaluation datasets are not comprehensive. The author only considers 55+302 articles among the daily evolving cyberspace, which simply yields a tiny set of CTI and cannot provide strong evaluation insights.

[1] ThreatKG: An AI-Powered System for Automated Open-Source Cyber Threat Intelligence Gathering and Management

**Questions:**

Please check the comments above.

---

> ### Author Response · Authors · 2025-11-19
>
> Thank you for the review.
>
> For the novelty: We acknowledge that chunking techniques have been used in prior work to process long texts. However, our main contribution is not the chunking method itself, but rather the systematic framework we have developed. This framework constrains and guides the behavior of LLM on CTI text through precise task definitions, a formal ontology, and a multi-stage pipeline. The core value of our work is demonstrating that, with this carefully designed framework, even locally deployed open-source models such as QWQ 32B can achieve and even surpass the state-of-the-art performance previously attainable only by large, commercial closed-source models like GPT-4o.
>
> On the comparison with ThreatKG: Thank you for mentioning ThreatKG. We believe our work is significantly different and has distinct novelty. First, ThreatKG is a system based on fine-tuned BERT models, which relies on extensive labeled data for training. In contrast, LLM4CTI is an LLM-based framework. This gives it an advantage that ThreatKG lacks: our approach can continuously evolve and improve simply by upgrading the backend LLM (from QWQ 32B to a more powerful future model) without any retraining. For instance, when a new model (like the current Qwen3 80B-A3B) becomes available, we can just swap the model to significantly improve the processing speed due to it uses less activation parameters to achieve similar performance like QWQ 32B. Second, ThreatKG has not publicly released its code or model, making a direct comparison impossible. In contrast, we have fully open-sourced our code and datasets to support community research.
>
> On the limitations of CTI articles: As we mentioned in related work, extracting CTI knowledge from structured sources (such as MITRE ATT&CK and NVD) has been extensively studied. However, these structured databases often provide only high-level summaries of threats. Unstructured CTI articles (such as blogs and security reports) provide greater detail about cyber threats. For example, consider the well-known CVE-2021-44228 (Log4Shell) vulnerability: the NVD database description for it is very brief. But extensive attack details, such as which malware exploited the vulnerability and the specific attack chains, are documented across various security blogs and reports, as shown in our Figure 1.
>
> On AI annotation and 15% noise: The "85% agreement" you mentioned does not mean our generated KG has 15% noise. This value was used to validate the reliability of our "LLM evaluation" method itself. Our evaluation process is as follows: (1) Our Ground Truth was manually reviewed and finalized by human experts. (2) During evaluation, we used Gemini 2.5 Pro as an automated judge to determine if our model's output matched the human ground truth. (3) The 85-87% agreement rate refers to the consistency between the AI judge's verdicts and the human judge's verdicts. This demonstrates that our evaluation method is reliable.
>
> On dataset size: Our 'Diverse CTI Dataset' contains 55 articles. In the cybersecurity domain, this scale is comparable to related work (such as MalKG and MalwareTextDB, which both used 64 articles).

---

> ### Author Response · Authors · 2025-11-23
>
> To verify whether LLM post-training can replace our prompt design, we trained Qwen3-4B-Instruct-2507 with SFT and RL under a one-pass setting, removing both chunking and dual-context processing. The RL reward combined two components: (1) a format reward, encouraging the model to generate step-by-step reasoning traces, and (2) a result reward, defined as the correctness of multiple-choice questions that Gemini 2.5 Pro automatically generated from 800 CTI articles and their corresponding auto-constructed knowledge graphs. Evaluation followed the same protocol as in our main paper, using the 55 manually annotated CTI articles as ground truth and Gemini 2.5 Pro as the semantic judge. We observed that after training, the 4B model developed step-by-step reasoning behavior but its language-generation stability degraded, and it sometimes failed to complete extraction within the 32k-token context window. Consequently, only 19 articles were successfully processed by all three settings within 32k tokens, and we report results on this overlap. On these 19 articles, the base Qwen3-4B achieved Precision 0.776, Recall 0.582, F1 0.614; Qwen3-4B with SFT achieved Precision 0.718, Recall 0.660, F1 0.640; and Qwen3-4B with SFT + RL achieved Precision 0.698, Recall 0.682, F1 0.632. RL increased recall but reduced precision, yielding no net F1 gain over SFT.
>
> Finally, in cost-sensitive and hardware-limited settings, there is an inherent trade-off between model size and context capacity. With our two RTX 6000 Ada GPUs, deploying larger models for inference requires shrinking the context window below 32k tokens, while full-scale training is practical only up to about 4B parameters with an 8k-token context. Therefore, post-training a one-pass model is not a substitute for our proposed LLM4CTI, which preserves long-range evidence and enables locally deployable open-source models to match or surpass commercial LLMs without relying on expensive cloud APIs or high-end hardware.

---

### Official Review · Reviewer_jx6C · 2025-11-03

**Soundness:** 3
**Presentation:** 3
**Contribution:** 2
**Rating:** 4
**Confidence:** 3

**Summary:**

The paper addresses the problem of translating Cyber Threat Intelligence (CTI) reports into knowledge graphs (KGs) using large language models (LLMs). It focuses on handling long CTI reports by dividing them into smaller chunks and proposes constraint on KG structure tailored specifically for CTI data. The empirical results show improvements over existing methods, although the gains are not very substantial.

**Strengths:**

- The problem is timely and important.

**Weaknesses:**

- While the proposed approach is effective, it appears to be mainly engineering-oriented, with limited novelty in methodological design.

- The paper lacks ablation studies to examine the contribution of each component in the proposed framework. In particular, it remains unclear how the length of CTI reports impacts the model’s effectiveness. It is possible that the observed improvements are due to other factors rather than the handling of long documents.

**Questions:**

- Provide ablation studies to investigate the effect on the CTI's length.

---

> ### Author Response · Authors · 2025-11-19
>
> Thank you for the review.
>
> For the novelty: We acknowledge that chunking techniques have been used in prior work to process long texts. However, our main contribution is not the chunking method itself, but rather the systematic framework we have developed. This framework constrains and guides the behavior of LLM on CTI text through precise task definitions, a formal ontology, and a multi-stage pipeline. The core value of our work is demonstrating that, with this carefully designed framework, even locally deployed open-source models such as QWQ 32B can achieve and even surpass the state-of-the-art performance previously attainable only by large, commercial closed-source models like GPT-4o.
>
> For the impact of CTI length: Regarding your question on how the length of CTI reports impacts effectiveness, we investigated the effect of length handling via an ablation study on chunk size in Appendix L and Table 6, by comparing 400, 200, and 100 tokens per chunk settings using the Qwen3 32B model. The results demonstrate that finer-grained processing, such as 100 tokens, can further improve the F1-score (from 78.53% to 82.30%). However, this improvement comes with a significant computational cost: processing with 100-token chunks takes approximately 4 times longer than with 400-token chunks. Additionally, the input for the final merger stage in the 100-token setting approaches the model's maximum context limit (32k tokens).
>
> For the ablation studies: To examine the contribution of each component, we conducted additional experiments using the QWQ 32B model to isolate the impact of our ontology design and dual-context input:
>
>     Impact of Ontology Design: We evaluated a baseline that uses only the standard definitions from the STIX 2.1 documentation, providing entity/relation types with just their standard one-sentence definitions, instead of our ontology which includes detailed definitions and examples. The results show a drop in performance: Recall drops from 78.66% (Ours) to 36.62% (Basic STIX).
>
>     Impact of Dual-Context & Chunking: We also tested a "One-Pass" baseline where we removed the dual-context and chunk-wise processing pipeline, feeding the full article directly into the model. The F1-score dropped from 81.58% to 70.53%, with Recall falling from 78.66% to 59.72%.
>
> Conclusion from Ablation Studies: These results confirm that smaller open-source models like QWQ 32B rely heavily on detailed prompt guidance to direct their workflow and explicit control over the context extraction process. Our framework provides the necessary guidance to enable them to perform complex extraction tasks that would otherwise fail.

---

> ### Author Response · Authors · 2025-11-23
>
> To directly test whether the LLM post-training can replace our dual-context, chunk-wise pipeline, we trained a small model (Qwen3-4B-Instruct-2507) with SFT and then RL. The RL reward combined two components: (1) a format reward, encouraging the model to generate step-by-step reasoning traces, and (2) a result reward, defined as the correctness of multiple-choice questions that Gemini 2.5 Pro automatically generated from 800 CTI articles and their corresponding auto-generated knowledge graphs. Evaluation followed the same protocol as in our main paper, using the 55 manually annotated CTI articles as ground truth and Gemini 2.5 Pro as the semantic judge. During inference, each article was processed in a single pass without chunking. We first observed that after training, the 4B model acquired step-by-step reasoning behavior, but its language-generation stability degraded, and at times it failed to complete the extraction within the 32k-token context window. Consequently, only 19 articles were successfully processed by all three settings within 32k tokens, and we report results on this overlap. On these 19 articles, the base Qwen3-4B achieved Precision 0.776, Recall 0.582, F1 0.614; Qwen3-4B with SFT achieved Precision 0.718, Recall 0.660, F1 0.640; and Qwen3-4B with SFT + RL achieved Precision 0.698, Recall 0.682, F1 0.632. RL increased recall but reduced precision, yielding no net F1 gain over SFT.
>
> Finally, in cost-sensitive and hardware-limited settings, there is an inherent trade-off between model size and context capacity. With our two RTX 6000 Ada GPUs, deploying larger models for inference requires shrinking the context window below 32k tokens, while full-scale training is practical only up to about 4B parameters with an 8k-token context. Therefore, post-training a one-pass model is not a substitute for our proposed LLM4CTI, which preserves long-range evidence and enables locally deployable open-source models to match or surpass commercial LLMs without relying on expensive cloud APIs or high-end hardware.

---

### Official Review · Reviewer_7oWs · 2025-11-04

**Soundness:** 2
**Presentation:** 2
**Contribution:** 3
**Rating:** 4
**Confidence:** 4

**Summary:**

This paper presents LLM4CTI, a system for automated CTI extraction and knowledge graph construction from long unstructured CTI articles.
The method attempts to address the challenges of extracting information from long CTI report/sources such as entities spread across the long content, irrelevant content. This is done by  splitting the content into chunks each processed by an LLM together with the full doc. as context. In addition, a GNN-based link prediction model is used to infer relationships among the entities.
The method was evaluated using a large CTI corpus and was compared with prior approaches, CTINexus, CTIKG, and GraphRAG in entity/relation extraction task. Results show high precision/recall using both proprietary and open-source LLMs.

**Strengths:**

1. Problem and motivation is clearly described
2. Evaluation was performed using a large dataset
3. Evaluating with local models, which shows good results
4. Demonstrating how the GNN-based link prediction can provide actionable value for analysts.

**Weaknesses:**

1. Related works is not informative and missing relevant papers, for example:
[a] Entity and relation extractions for threat intelligence knowledge graphs
[b] KnowCTI: Knowledge-based cyber threat intelligence entity and relation extraction
[c] CyberEntRel: Joint extraction of cyber entities and relations using deep learning
[d] CyberKG: Constructing a Cybersecurity Knowledge Graph Based on SecureBERT_Plus for CTI Reports
[e] Ctinexus: Automatic cyber threat intelligence knowledge graph construction using large language models
and there are more...

2. Limited novelty - basically applying a common practice of splitting the input to an LLM into small sections to improve its performance.

3. An evaluation of removing important component of the method such as chunk-wise processing will help in understanding the effectiveness of the method. Also, what if removing the GNN? the dual-context...

4. Figure 1 is hard to read and should be improved.

5. Table 1 - the text is too small and hard to read.

6. Need to discuss and evaluate the robustness of LLM4CTI to prompt injection attacks.

**Questions:**

1. How do you address images within the CTI, some are relevant and some are not?
2. You are relying on prompts and LLMs. How robust is your system?
3. How was the dataset labeled and annotated?
4. Can you provide details on the average execution time?

---

> ### Author Response · Authors · 2025-11-19
>
> Thank you for your careful review and valuable feedback.
>
> For the related work: Thank you for pointing out several related papers. For works such as KnowCTI, CyberEntRel, and CyberKG, we noticed that their model weights are not available for reproduction. Therefore, in our study we selected frameworks that are publicly available and reproducible, such as GraphRAG and CTINexus, as our primary baselines.
>
> For the figure and table readability: We appreciate the reviewer’s comment regarding Figure 1 and Table 1. In the revised version, we will present them in a landscape (rotated 90 degrees) layout with enlarged fonts so that all values remain legible.
>
> For the novelty: We acknowledge that chunking techniques have been used in prior work to process long texts. However, our main contribution is not the chunking method itself, but rather the systematic framework we have developed. This framework constrains and guides the behavior of LLM on CTI text through precise task definitions, a formal ontology, and a multi-stage pipeline. The core value of our work is demonstrating that, with this carefully designed framework, even locally deployed open-source models such as QWQ 32B can achieve and even surpass the state-of-the-art performance previously attainable only by large, commercial closed-source models like GPT-4o.
>
> For the prompt injection: LLM4CTI is designed to process CTI reports from professional, trustworthy security websites like Securitylist. A successful prompt-injection attack would require compromising such source websites to plant adversarial instructions. We consider this scenario to be outside the robustness scope of our system.
>
> For the images: At present, our method (and all baselines such as CTINexus, GraphRAG, CTIKG) focuses on unstructured text and does not process embedded images in CTI articles. We agree this is an important direction. In future work, we plan to incorporate VLM and OCR to extract and fuse information from images.
>
> For the dataset annotation: As described in Appendix F, our ground truth was generated as follows: we first used Gemini 2.5 Pro to produce an initial sketch, then a PhD-level human expert conducted thorough manual review and correction for each article-level graph.
>
> For the average runtime: On our RTX 6000 Ada GPU, the QWQ 32B model requires about 120 seconds on average to process an article. When using the o4-mini API, OpenAI’s token generation rate is variable, and large batches can leverage parallelism. Therefore, we view token usage as a more stable and reasonable metric than time.

---

> ### Author Response · Authors · 2025-11-23
>
> To directly test whether the LLM post-training can replace our dual-context, chunk-wise pipeline, we trained a small model (Qwen3-4B-Instruct-2507) with SFT and then RL. The RL reward combined two components: (1) a format reward, encouraging the model to generate step-by-step reasoning traces, and (2) a result reward, defined as the correctness of multiple-choice questions that Gemini 2.5 Pro automatically generated from 800 CTI articles and their corresponding auto-generated knowledge graphs. Evaluation followed the same protocol as in our main paper, using the 55 manually annotated CTI articles as ground truth and Gemini 2.5 Pro as the semantic judge. During inference, each article was processed in a single pass without chunking. We first observed that after training, the 4B model acquired step-by-step reasoning behavior, but its language-generation stability degraded, and at times it failed to complete the extraction within the 32k-token context window. Consequently, only 19 articles were successfully processed by all three settings within 32k tokens, and we report results on this overlap. On these 19 articles, the base Qwen3-4B achieved Precision 0.776, Recall 0.582, F1 0.614; Qwen3-4B with SFT achieved Precision 0.718, Recall 0.660, F1 0.640; and Qwen3-4B with SFT + RL achieved Precision 0.698, Recall 0.682, F1 0.632. RL increased recall but reduced precision, yielding no net F1 gain over SFT.
>
> Finally, in cost-sensitive and hardware-limited settings, there is an inherent trade-off between model size and context capacity. With our two RTX 6000 Ada GPUs, deploying larger models for inference requires shrinking the context window below 32k tokens, while full-scale training is practical only up to about 4B parameters with an 8k-token context. Therefore, post-training a one-pass model is not a substitute for our proposed LLM4CTI, which preserves long-range evidence and enables locally deployable open-source models to match or surpass commercial LLMs without relying on expensive cloud APIs or high-end hardware.

---

### Meta-Review · Area_Chair_TMc3 · 2025-12-10

**Summary:**

The paper studies extracting cyber threat intelligence from long CTI articles using a dual-context, chunk-wise framework and a GNN for link prediction.

Reviewers find the problem important and the general idea reasonable. They agree the method is clearly described and the experiments are conducted on real CTI data. However, all reviewers also point out concerns about novelty. They believe the approach is mainly engineering-oriented and similar ideas have been explored in prior work. Several reviewers highlight missing ablation studies, unclear impact of CTI length, and the need to isolate the contribution of each component.

One reviewer strongly questions dataset size, annotation quality, and comparison with related work. Reviewers also raise fairness concerns regarding baseline choices and ask for deeper analysis.

The paper does not reach the bar of ICLR.

**Reviewer Concerns:**

Aall reviewers point out concerns about novelty. They believe the approach is mainly engineering-oriented and similar ideas have been explored in prior work. Several reviewers highlight missing ablation studies, unclear impact of CTI length, and the need to isolate the contribution of each component.

One reviewer strongly questions dataset size, annotation quality, and comparison with related work. Reviewers also raise fairness concerns regarding baseline choices and ask for deeper analysis.

**Reviewer Scores:**

The reviewers are unlikely to change their scores due to the lack of novelty.

---

### Decision · Program_Chairs · 2026-01-26

Reject